# An Image Analysis for the Development of a Skin Change-Based AI Screening Model as an Alternative to the Bite Pressure Test

**DOI:** 10.3390/healthcare13080936

**Published:** 2025-04-18

**Authors:** Yoshihiro Takeda, Kanetaka Yamaguchi, Naoto Takahashi, Yasuhiro Nakanishi, Morio Ochi

**Affiliations:** 1Division of Fixed Prosthodontics and Oral Implantology, Department of Oral Rehabilitation, School of Dentistry, Health Sciences University of Hokkaido, Tobetsu 061-0293, Japan; take3738@hoku-iryo-u.ac.jp (Y.T.); k-yamaguchi@hoku-iryo-u.ac.jp (K.Y.); nakanisi@hoku-iryo-u.ac.jp (Y.N.); 2Advanced Intelligence Technology Center, Sapporo City University, Sapporo 060-0061, Japan; na.takahashi@scu.ac.jp

**Keywords:** CNN, transfer learning, masticatory function, occlusal pressure

## Abstract

**Background/Objectives:** Oral function assessments in hospitals and nursing facilities are mainly performed by nurses and caregivers but are sometimes not properly assessed. As a result, elderly people are not provided with meals appropriate for their masticatory function, increasing the risk of aspiration and other complications. In the present study, we aimed to examine image analysis conditions in order to create an AI model that can easily and objectively screen masticatory function based on occlusal pressure. **Methods:** Sampling was conducted at the Hokkaido University of Health Sciences (Hokkaido, Japan) and the university’s affiliated dental clinic in Hokkaido. **Results:** We collected 241 waveform images of changes in skin shape during chewing over a 20 s test period from 110 participants. Our study used two approaches for image analysis: convolutional neural networks (CNNs) and transfer learning. In the transfer learning analysis, MobileNetV2 and Xception achieved the highest classification accuracy (validation accuracy: 0.673). **Conclusions:** Therefore, it was determined that analyses of waveform images of changes in skin shape may contribute to the development of a skin change-based screening model as an alternative to the bite pressure test.

## 1. Introduction

Masticatory function evaluations consist of subjective and objective assessments and have been widely used in Japan. The Chewing Efficiency Determination Table [1] and the Basic Checklist [2] are used as subjective evaluations. The chewing efficiency evaluation [3] and occlusal pressure [4] are used as objective evaluations. These objective evaluations are also used in the diagnosis of oral dysfunction. Several studies have reported the cut-off points and correlations between masticatory force and chewing ability and between tongue pressure and tongue–lip motor function [5]. Furthermore, decreases in occlusal force and masticatory dysfunction are both highly indicative of oral dysfunction [6], and occlusal pressure is related to the activity of the masticatory muscles [7,8]. Therefore, it is desirable to include occlusal pressure test results in the evaluation of masticatory function.

The discrepancy between subjective and objective assessments increases under conditions of a low number of teeth [9,10]. Regardless of whether patients evaluate their chewing function as good, if an objective evaluation determines that their chewing function is impaired, then they are at a higher risk of malnutrition and dementia [11].

Therefore, it is necessary to appropriately evaluate masticatory function while paying attention to the discrepancy between subjective and objective evaluations; however, this requires specialized knowledge regarding masticatory function.

Conversely, oral function assessments in hospitals and nursing facilities are mainly performed by nurses and caregivers, but they are often conducted using subjective assessments, such as the Basic Checklist [12], and objective assessments can only be performed by those with previous experience [13].

It has been reported that 35% of elderly people living in facilities and 68% of elderly people receiving home care consume foods that are incompatible with their masticatory function, and they are not provided with meals appropriate for their masticatory function [12].

Therefore, it is important to develop a method that allows nurses and caregivers to easily screen masticatory function based on occlusal pressure without the need for special equipment or dental expertise. It has been reported that there is a relationship between maximum bite force and masseter muscle thickness in healthy elderly people [14]. Furthermore, studies have demonstrated a relationship between the preferred side of chewing and dynamic occlusion parameters [15]. However, so far, there is no evidence showing a correlation between bite force and changes in skin morphology. Moreover, after adjusting for occlusion with the preferred chewing side, the duration of the chewing cycle, and the mean difference vector value of the chewing path, the movement of the facial skin during chewing in healthy subjects was stable on both sides [16]. Thus, we adopted skin changes as a surrogate indicator of bite pressure and considered developing an artificial intelligence (AI)-based bite pressure assessment. AI-based assessment has become a widely used technology in the dental field [17,18], with numerous applications in the differentiation and diagnosis of dental diseases and in predicting treatment outcomes [19,20]. Although oral health management systems using AI [21] and swallowing function assessments [22] have been reported, there have been no reports on the evaluation of masticatory function using AI based on occlusal pressure.

The aim of this study was to examine image analysis conditions for creating an AI model that can easily and objectively screen masticatory function based on occlusal pressure.

## 2. Materials and Methods

### 2.1. Subjects and Recruitment Method

The subjects of this study were students enrolled at the Hokkaido University of Health Sciences and the university’s affiliated dental clinic in Hokkaido, Japan. We explained the purpose of the study in advance, and the subjects provided consent for participation in the study. To minimize the influences of occlusal support and the current number of teeth [23], we sampled participants who had an occlusal support zone corresponding to Eichner A. The exclusion criterion was the presence of symptoms of temporomandibular joint disorder.

### 2.2. Investigators and Ethical Considerations

One dentist from the Health Sciences University of Hokkaido conducted data sampling at the Hokkaido University of Health Sciences and the university’s affiliated dental clinic in Hokkaido, Japan. On the day of the survey, the subjects were given a consent withdrawal form and were informed that they could withdraw their consent at any time, and their intention to participate in the study was reconfirmed. Furthermore, this study was conducted with the approval of the ethical review board of the Health Sciences University of Hokkaido (approval number: 208).

### 2.3. Test Items

The occlusal pressure test was performed using the Dental Pressure Scale II^®^ (GC, Tokyo, Japan,) (Figure 1), which subjects were asked to insert into their mouth and bite down on for 3 s. After clenching, occlusal pressure was measured using a dedicated measuring instrument. The participants practiced clenching using a protective sheet in advance. The reference value was set to 500 N, based on the criteria for oral dysfunction [24]. Mastication motion was measured using bitescan^®^ (BH-BS1RR, SHARP, Tokyo, Japan). Bitescan^®^ is a device with an infrared distance sensor and an accelerometer, and it has been shown to be effective in measuring the number of chews [25]. Bitescan^®^ can measure changes in the skin surface behind the ear and provide information on chewing frequency, chewing tempo, the number of chews per mouthful, and head position (Figure 2). The device is designed to be worn on the right ear and includes an adjustable ear hook. The appropriate ear hook for each participant was selected based on the size of their ear. Prior to measurement, the subjects wore bitescan^®^ and connected it to a tablet device (Sharp SH-T01) via Bluetooth. We then confirmed that data could be properly collected using a dedicated application. For data collection with bitescan^®^, gummy jelly was used as the test food in accordance with the mastication function testing methods. Prior to the test, the participants practiced chewing the gummy jelly on both sides to ensure that the calibration feature of bitescan^®^ could sample data correctly. Changes in skin shape during chewing over the 20 s test period were plotted by bitescan^®^ as waveforms, with time (0.3 s/scale) on the horizontal axis and distances (Hz) on the vertical axis (Figure 3).

### 2.4. Calibration

Calibration was performed for both the examiner and the participant. The examiner confirmed the procedures and followed the data sampling protocol. To standardize posture during data sampling, the participants were seated in a chair with a backrest.

## 3. Image Analysis

In the field of AI image analysis, a system is trained to accurately extract morphological features from image data by reading in the data and iteratively learning from it. Image data obtained from Bitescan^®^ were linked to the results of the bite pressure test and classified (Figure 4). Additionally, image data related to masticatory function test results were aggregated, and an algorithm was developed to achieve a training-to-validation data sampling ratio of 80%:20%. The analysis process involves the iterative training of the AI system to accurately extract morphological features from the image data. Initially, a model for image classification was constructed using the training datasets, and its performance was evaluated based on accuracy and loss metrics. Subsequently, validation data were introduced to assess validation accuracy and validation loss, completing one learning cycle, commonly referred to as an “epoch”. In subsequent epochs, the model was iteratively refined to further minimize training data loss, followed by re-evaluation using validation data. By repeating this cycle, a high-performance model with optimal discriminative ability was developed. However, as the number of epochs increases, the risk of overfitting emerges. Overfitting, a phenomenon where the model becomes excessively tailored to the training data, compromises its generalization performance on unseen data. This issue can be identified by a significant increase in validation loss, indicating that the model’s adaptability to new data has deteriorated. To mitigate overfitting, an algorithm was implemented to monitor fluctuations in validation loss during the training process. The final model was selected at the optimal juncture, defined as the point at which there was a validation loss.

Our study employed two approaches for the image analysis: convolutional neural networks (CNNs) and transfer learning. A CNN is a type of deep neural network with multiple layers, where each layer learns specific features of the target image. The network consists of three main layers: convolutional layers, pooling layers, and fully connected layers [26].

Transfer learning is a technique in which a pre-trained model, trained on a large dataset, is fine-tuned by retraining only its output layer [27]. This method can significantly reduce the training time and achieve good learning outcomes, even with a small dataset. Several models are available for transfer learning, and six models were used in this study: VGG16, ResNet50, DenseNet121, InceptionV3, Xception, and MobileNetV2. VGG16 is a 16-layer CNN model pre-trained on the large-scale image dataset ImageNet, and it was developed by the VGG team at the University of Oxford. ResNet50 is a 50-layer CNN model pre-trained on ImageNet. By utilizing multiple layers, this model achieves improved accuracy compared to shallower networks. ResNet models were designed to mitigate the vanishing gradient problem, allowing for deeper layers to achieve better performance. DenseNet121 is a 121-layer CNN model pre-trained on ImageNet. This model is an improvement over the ResNet model. InceptionV3 is a 48-layer CNN model pre-trained on ImageNet and developed by Google. It was designed for image classification tasks and can classify images into 1000 different categories. Xception is a 71-layer CNN model pre-trained on ImageNet. It is an improvement over the Inception model. MobileNetV2 is a 53-layer CNN model pre-trained on ImageNet. It is an improvement over MobileNetV1 and designed for mobile and edge devices. For the image analysis, we used Python (version 3.9.0) as the programming language, and we used TensorFlow (version 2.10.0) and Keras (version 2.10.0) as the deep learning libraries. Analyst: Prof. Naoto Takahashi, Director of the Advanced Intelligence Technology Center, Sapporo City University, Japan.

### Setting Image Processing Conditions

Before the image analysis, the optimal image processing conditions were determined by comparing two conditions for the vertical axis of the chewing data: (1) the minimum and maximum values during the 20 s chewing period were fixed to the range of 0 to 700, which included the minimum and maximum values of all sampled images (Figure 5), and (2) the axis range was set to the minimum and maximum values during the 20 s chewing period for each image (Figure 6). The learning curves obtained from both conditions were compared, and the optimal image processing parameters for analysis were selected. Analyst: Prof. Naoto Takahashi, Director of the Advanced Intelligence Technology Center, Sapporo City University, Japan.

## 4. Results

This study included 110 participants, comprising 66 males and 44 females. The mean age of the participants was 24.8 years, with a standard deviation of 4.7. No participants met the exclusion criteria. Data were collected from each participant two to three times, resulting in 241 image samples. Based on the results of the bite pressure test, the image samples were assigned to two groups: a normal bite pressure group (150 samples) and a reduced bite pressure group (91 samples).

When the minimum and maximum values on the vertical axis of the obtained image data were used, the accuracy of the learning curve improved from the 20th learning epoch (Figure 7). However, when the minimum and maximum values on the vertical axis were unified to the range of 0 to 700, almost no change was observed in the learning curve, even with an increase in the number of learning epochs (Figure 8).

As shown in Figure 7, as the number of learning epochs increased, the accuracy improved, and the loss decreased. However, although the validation accuracy improved, the validation loss showed the opposite trend, suggesting that overfitting occurred in the CNN analysis on this dataset.

In the transfer learning analysis, we compared the classification accuracy and learning curve of each model. The epochs with the lowest model accuracy (validation accuracy) and validation loss were as follows: Xception: 0.673 (epoch 30, Figure 9); DenseNet121: 0.612 (epoch 13, Figure 10); InceptionV3: 0.449 (epoch 27, Figure 11); MobileNetV2: 0.673 (epoch 26, Figure 12); ResNet50: 0.612 (epoch 42, Figure 13); and VGG16: 0.633 (epoch 42, Figure 14). From these results, it was found that MobileNetV2 and Xception achieved the highest classification accuracy.

## 5. Consideration

In our study, the normal occlusal pressure group contained significantly more males, which was consistent with the findings of [7]. The measurement sites of Bitescan^®^ did not include the attachments of the origins and insertions of the masticatory muscles, which was thought to simplify complex jaw movement data. Upon examining the optimal image processing conditions, a comparison was made between fixing the vertical axis to the range of 0–700 and setting the minimum–maximum values of the distance for each data. A change in the learning curve was observed when the vertical axis was set to the minimum–maximum values of the distance. This suggests that the latter condition was more appropriate for the dataset in this study (Figure 7). The reason why the learning curve became unclear in the former condition is likely because, with the minimum and maximum values fixed, the images from the two groups became more similar, making it more difficult for the AI to extract features from the images (Figure 8). Furthermore, it was found that, by using transfer learning, a CNN method, it was possible to construct a model that can classify normal and reduced occlusal pressure with an accuracy of approximately 67.3% based on bitescan^®^ image data. Four strengths of the present study are as follows: First, the bitescan^®^ image data used in this study differ from those used in previous studies in that they visualize skin changes during chewing. In addition, the accuracy of the learning curve based on the bitescan^®^ image data improved as the number of epochs increased, and this is expected to be useful for CNN analyses and transfer learning. Second, it was found that setting the distance of the image data in the minimum to maximum range was effective for image analysis. Third, it was possible to classify normal occlusal pressure and reduced occlusal pressure with an accuracy of approximately 67.3% by applying transfer learning. In transfer learning, the accuracy varied across the six models, and the number of learning iterations required to achieve that accuracy also differed. In this study, MobileNetV2 and Xception demonstrated high accuracy values, suggesting that, although the learning curve was unclear, applying transfer learning with these two models may be an effective approach. Fourth, bite pressure testing should be conducted exclusively by dental professionals. In contrast, bitescan^®^ is not classified as a medical device and can be operated without specialized dental knowledge. It is also almost non-invasive to the patient, and there is no need to set aside additional time for testing. Its test results are presented in a clear and straight forward manner, making it highly feasible for practical implementation. Moreover, it has been effectively utilized in public health education programs in Fukuoka and Niigata, Japan. Therefore, our study can suggest that bitescan^®^ can help nurses and caregivers to easily screen masticatory function based on occlusal pressure without the need for dental expertise.

Three limitations of the present study are as follows: First, the sample size was small, and methods such as image augmentation were not used to increase the sample size. Furthermore, due to the small sample size, the learning curve showed overfitting characteristics, and the discrimination accuracy was approximately 67.3%. Using a deep learning model with X-ray imaging, implant size classification achieved a performance score of 90% or higher [28] and diagnostic accuracy for caries was reported as 89.0% (80.4–93.3) for the premolar model, 88.0% (79.2–93.1) for the molar model, and 82.0% (75.5–87.1) for the combined premolar and molar models [29]. The accuracy of our study was lower than that reported in previous studies, due to the smaller dataset used in this study compared to others. Additionally, a scoping review [30] evaluating the efficacy of deep learning models in tasks involving periodontitis and oral implants found that only a limited number of studies (*n* = 7) had a low risk of bias, underscoring the need for caution when directly comparing the results with prior research. Second, the participants were limited to young people with occlusal support areas in the Eichner A group to minimize the influences of occlusal support and the current number of teeth [23], the findings of our study cannot be directly applied to older people. Third, skin changes also may be underestimated due to decreased skin tension and skin wrinkles in elderly patients. As chewing function evaluation is one of the important factors in determining the appropriate form of food intake [12], in future research, we would like to expand the range of data collection beyond occlusal support sites and age restrictions, increase the number of samples to incorporate parameters such as the reduced number of teeth and the consequent variations in masticatory patterns in elderly populations, and improve the analytical accuracy of the constructed AI model, with the aim of building a system that can perform simple functional evaluations based on occlusal pressure in nursing and care settings.

## 6. Conclusions

Despite the limitations of this study, it was found that, by performing an image analysis under the conditions set in this study, it was possible to classify occlusal pressure from the shape changes in the skin during chewing with an accuracy of about 67.3%.

## Figures and Tables

**Figure 1 healthcare-13-00936-f001:**
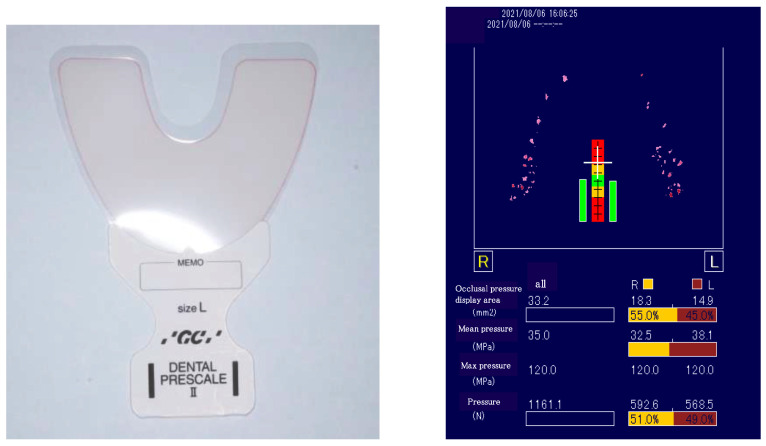
Occlusal pressure test.

**Figure 2 healthcare-13-00936-f002:**
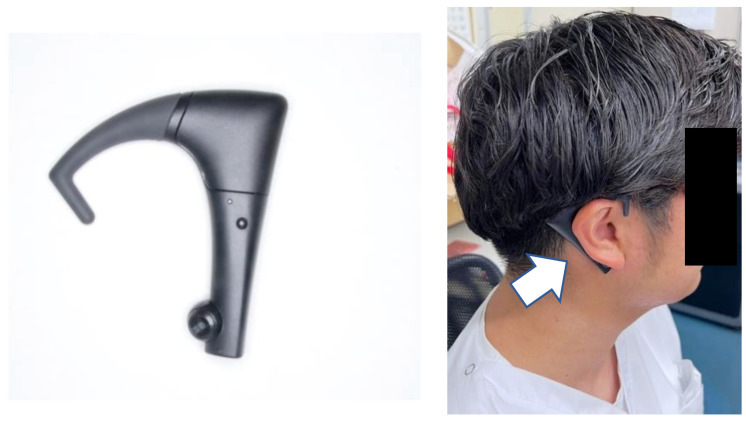
Bitescan^®^.

**Figure 3 healthcare-13-00936-f003:**
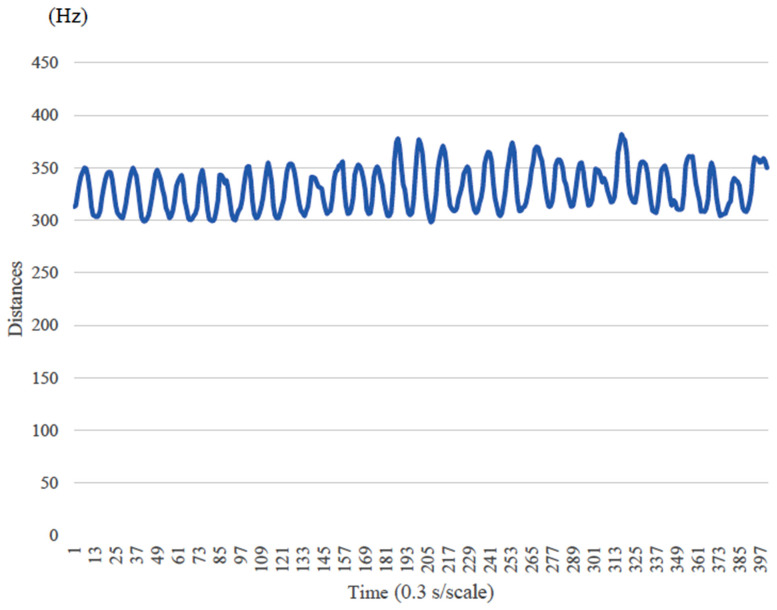
Waveform (vertical axis: distances, horizontal axis: time).

**Figure 4 healthcare-13-00936-f004:**
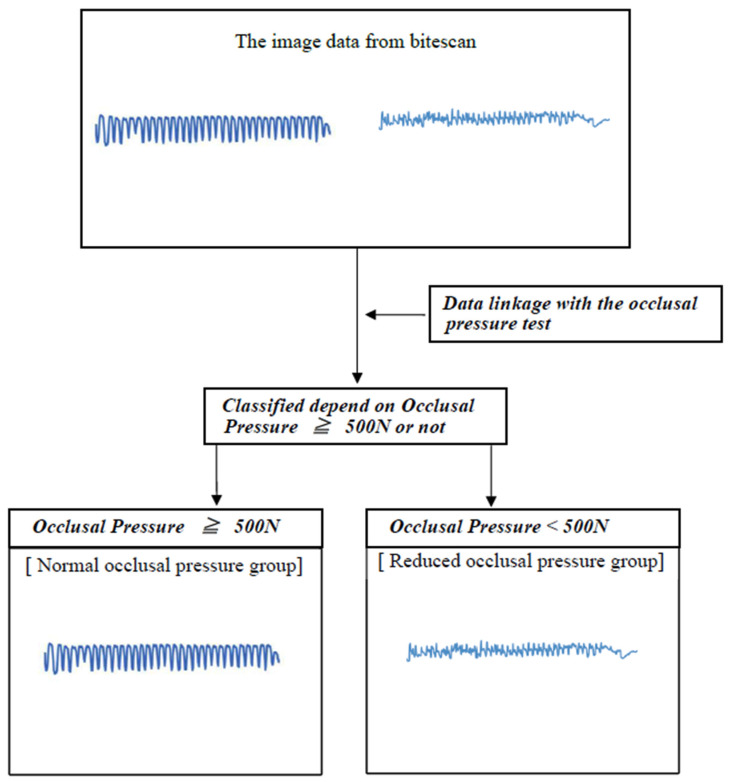
Classification method. Analyst: Prof. Naoto Takahashi, Director of Advanced Intelligence Technology Center, Sapporo City University, Japan.

**Figure 5 healthcare-13-00936-f005:**
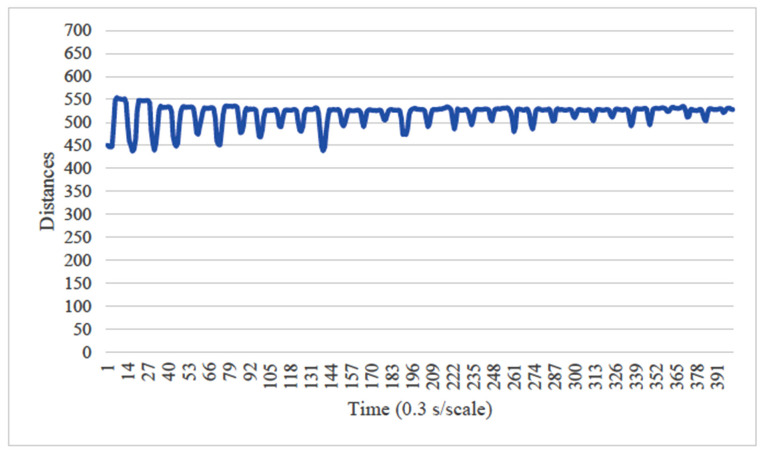
Image in case of setting vertical axis fixed to range of 0 to 700 Hz. Analyst: Prof. Naoto Takahashi, Director of Advanced Intelligence Technology Center, Sapporo City University, Japan.

**Figure 6 healthcare-13-00936-f006:**
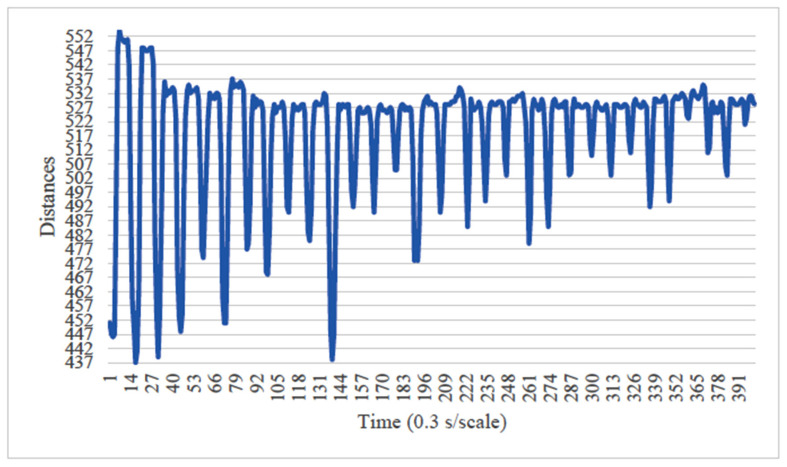
An image unified in the range from the minimum value to the maximum value of each datapoint. Analyst: Prof. Naoto Takahashi, Director of the Advanced Intelligence Technology Center, Sapporo City University, Japan.

**Figure 7 healthcare-13-00936-f007:**
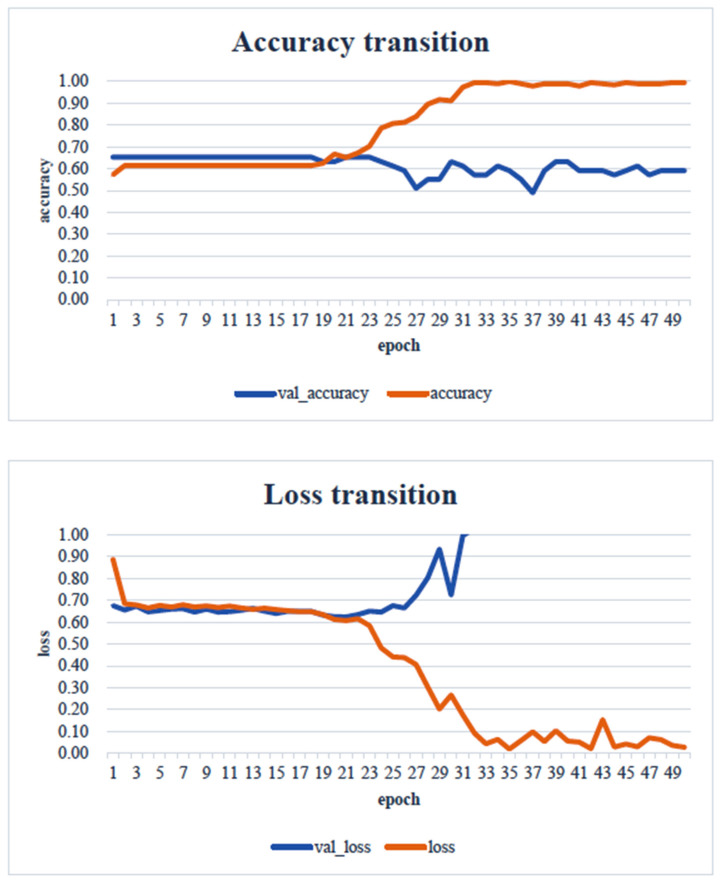
Learning curve: an image unified in the range from the minimum value to the maximum value of each datapoint. Analyst: Prof. Naoto Takahashi, Director of the Advanced Intelligence Technology Center, Sapporo City University, Japan.

**Figure 8 healthcare-13-00936-f008:**
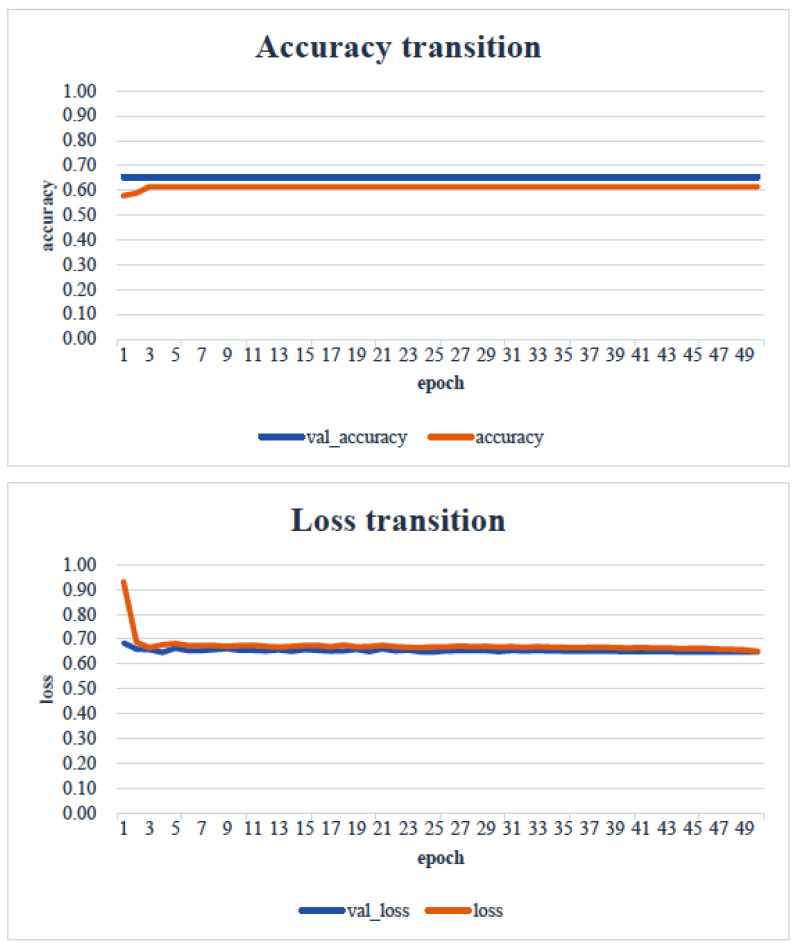
Learning curve: image showing range from 0 to 700 Hz. Analyst: Prof. Naoto Takahashi, Director of Advanced Intelligence Technology Center, Sapporo City University, Japan.

**Figure 9 healthcare-13-00936-f009:**
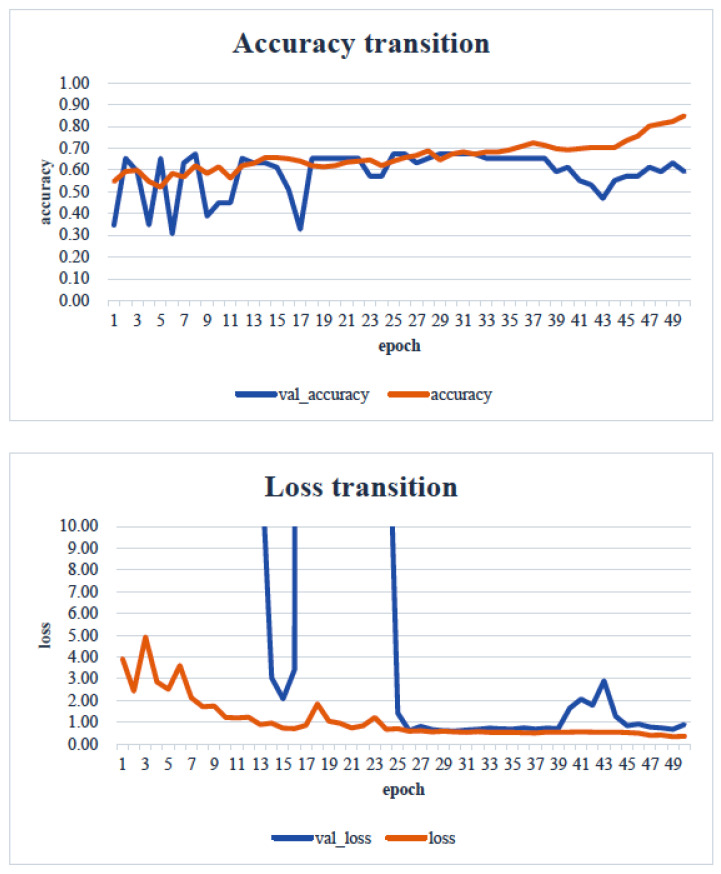
Transfer learning: Xception. Analyst: Prof. Naoto Takahashi, Director of Advanced Intelligence Technology Center, Sapporo City University, Japan.

**Figure 10 healthcare-13-00936-f010:**
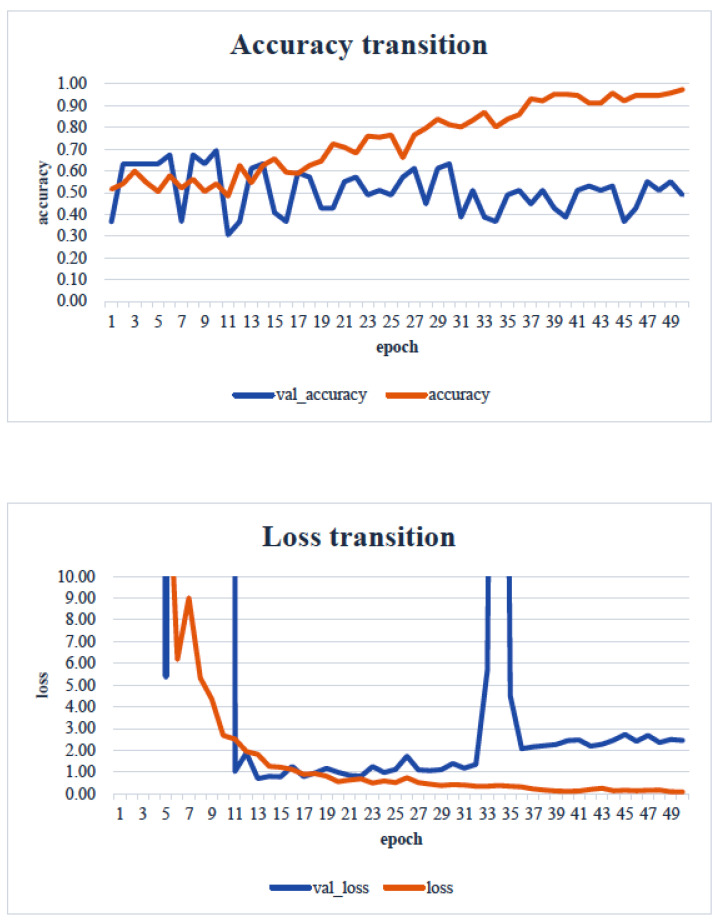
Transfer learning: Densenet121. Analyst: Prof. Naoto Takahashi, Director of Advanced Intelligence Technology Center, Sapporo City University, Japan.

**Figure 11 healthcare-13-00936-f011:**
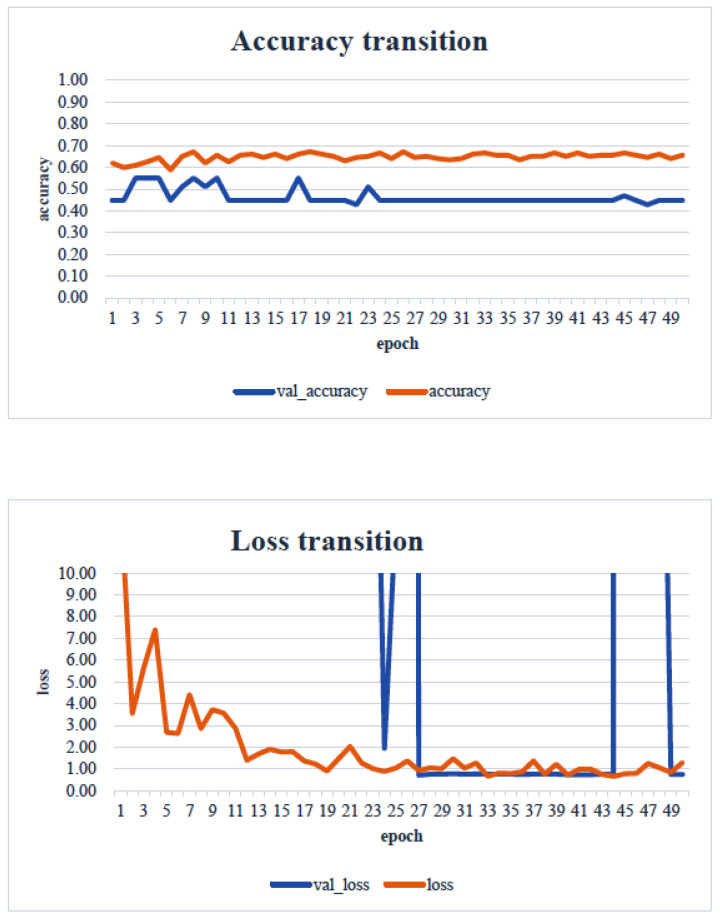
Transfer learning: InceptionV3. Analyst: Prof. Naoto Takahashi, Director of Advanced Intelligence Technology Center, Sapporo City University, Japan.

**Figure 12 healthcare-13-00936-f012:**
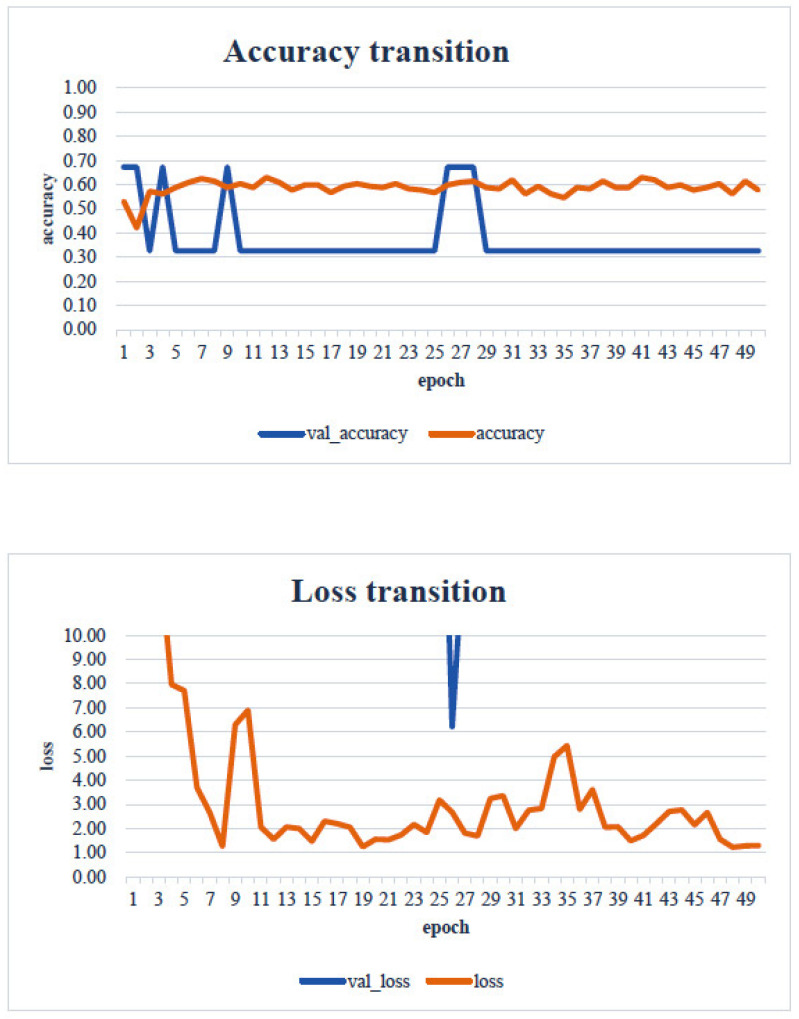
Transfer learning: MobileNetV2. Analyst: Prof. Naoto Takahashi, Director of Advanced Intelligence Technology Center, Sapporo City University, Japan.

**Figure 13 healthcare-13-00936-f013:**
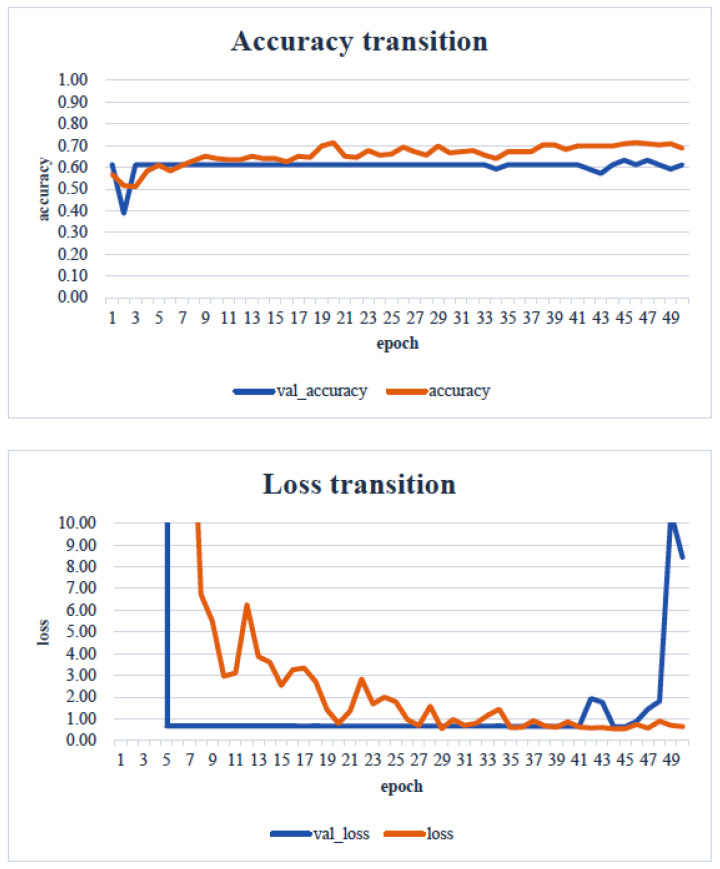
Transfer learning: ResNet50. Analyst: Prof. Naoto Takahashi, Director of Advanced Intelligence Technology Center, Sapporo City University, Japan.

**Figure 14 healthcare-13-00936-f014:**
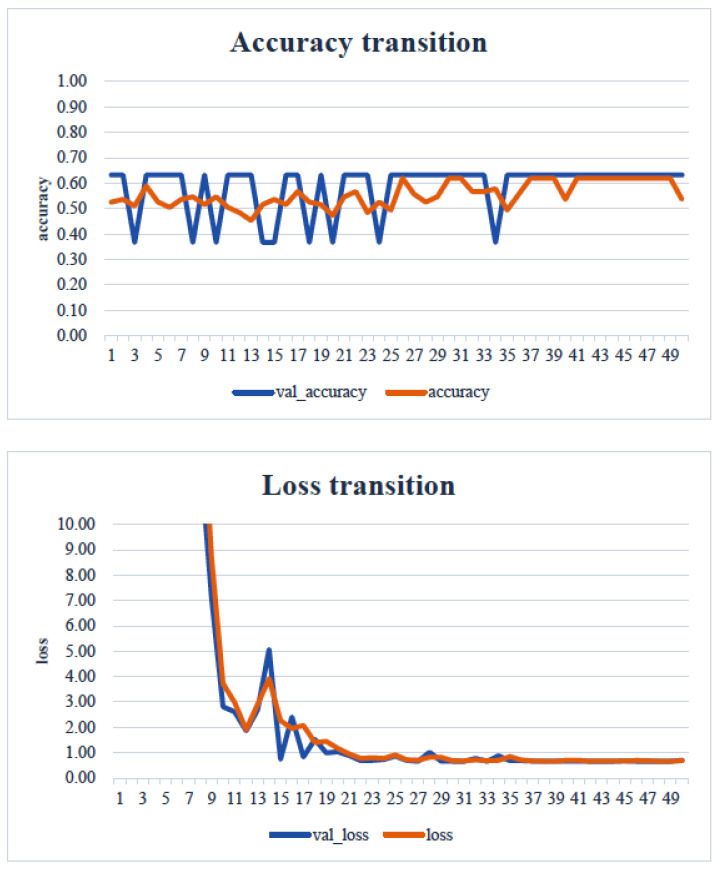
Transfer learning: VGG16. Analyst: Prof. Naoto Takahashi, Director of Advanced Intelligence Technology Center, Sapporo City University, Japan.

## Data Availability

Obtained from the participants.

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
