# Peer review of "An Image Analysis for the Development of a Skin Change-Based AI Screening Model as an Alternative to the Bite Pressure Test"

_healthcare, 2025, doi:10.3390/healthcare13080936_

Round 1

Reviewer 1 Report

Comments and Suggestions for Authors

This study presents an innovative approach to assessing masticatory function using image analysis and AI modeling. The proposed method, leveraging convolutional neural networks (CNNs) and transfer learning, offers a non-invasive and objective alternative to the traditional bite pressure test. The research is well-structured, with clearly defined objectives, methodology, and results. The purpose is to contribute to oral healthcare, especially for older adults. However, some areas require further refinement to enhance their clinical impact and applicability.

Strengths

  • The study addresses a significant clinical gap by providing an objective method for evaluating masticatory function, particularly relevant for elderly individuals in hospitals and nursing homes.
  • AI-driven image analysis represents a modern and potentially more accessible alternative to conventional bite pressure tests, reducing reliance on specialized dental equipment.
  • The study includes a substantial sample size (110 participants, 241 waveform images), which adds credibility to the findings.
  • Using CNNs and transfer learning (MobileNetV2 and Xception) enhances the robustness of the analysis by leveraging state-of-the-art deep learning models.

Weakness

  • The achieved validation accuracy of 0.673 suggests that the method has potential for clinical application. However, the study was conducted on young adults, while its intended application is for older adults. The authors should discuss potential adaptations to improve the model’s effectiveness in elderly populations.
  • Including comparisons with other AI-based models in the same field would help contextualize the model’s performance and identify areas for further optimization.
  • While the study introduces an alternative to the bite pressure test, a more comprehensive discussion on real-world applications is necessary. How feasible is the implementation of this screening tool in nursing homes and healthcare facilities? Would additional training be required for caregivers? Considering cost-effectiveness, usability, and patient compliance would strengthen the study’s practical contributions.

Conclusion

This study provides a promising foundation for AI-based masticatory function screening, demonstrating the feasibility of a viable alternative to traditional bite pressure tests. However, further refinements in model accuracy and validation studies on elderly populations are essential for real-world clinical implementation. Expanding the discussion on practical deployment would further enhance the study’s relevance and impact.

Comments on the Quality of English Language

Minor editing.

Reviewer 2 Report

Comments and Suggestions for Authors
  1. Title – Include the term “AI” or “Digital” in the title to increase article impact.
  2. Abstract – In the background, include a sentence as to why oral function assessments among elderly are needed in clinical practice, and how it relates to masticatory function.
  3. The title mentions a “skin-change-based AI model”. However, nothing is mentioned about skin-change in response to occlusal pressure in the introduction. This has to be elaborated by the authors.
  4. It is interesting that the authors used a device to measure skin surface movements in the retromandibular area based on infra-red and accelerometer sensors. While the study envisions this method for assessment of occlusal pressure in elderly patients, the study population for evaluation has been young dental school volunteers (mean age 24.8 years). This seems paradoxical as the population of the study is not age matched for the target population aimed for outcome analysis. Authors should explain about this and also state about it as a limitation.
  5. How would the authors factor in for loss of skin turgor and skin wrinkling in elderly patients while evaluating with the “Bitescan” device? This also has to be explained clearly.
  6. Post-calibration, what was the measure of reliability (Kappa score). Mention this.
  7. Please check and correct the legend for all figures. Where AI models/programs are mentioned in the legend, the developer credentials should be included in brackets.
  8. Lines 139-154 - the developer credentials for AI models/programs, like company name, version and country of origin, should be included in brackets.
  9. Authors must include, in the results, more detailed interpretations and inferences for the figures 7 – 14 (Describing AI model-based outputs). This would be imperative because majority of the journal readers would not be fluent with AI methodology and terminologies.
  10. Conclusion – An accuracy of 67.3% may be higher, but from a clinical and epidemiological standpoint, isn’t this still too low? Would you prescribe a treatment or diagnostic approach with just 2/3rd accuracy to yourself or your patient? This has to be discussed in detail, with ways of overcoming. Otherwise, the entire outcome of this report would be under question.

Round 2

Reviewer 1 Report

Comments and Suggestions for Authors

Congratulations!

Comments on the Quality of English Language

Minor English editing.

Reviewer 2 Report

Comments and Suggestions for Authors

Thanks for revising the manuscript based on the review comments.

All the very best.